# Sources of Carotenoids in Amazonian Fruits

**DOI:** 10.3390/molecules29102190

**Published:** 2024-05-08

**Authors:** Orquidea Vasconcelos dos Santos, Rosely Carvalho do Rosário, Barbara E. Teixeira-Costa

**Affiliations:** 1Graduate Program in Food Science and Technology (PPGCTA), Technology Institute, Federal University of Pará, Belém 66075-110, Pará, Brazil; rosely@ufpa.br; 2Institute of Health Sciences, Faculty of Nutrition, Federal University of Pará, Belém 66075-110, Pará, Brazil; betcosta@ufam.edu.br; 3Graduate Program in Biotechnology (PPGBIOTEC), Federal University of Amazonas, Manaus 69067-005, Amazonas, Brazil; 4Department of Nutrition and Dietetics, Faculty of Nutrition Emília de Jesus Ferreiro, Federal Fluminense University, Niterói 24020-140, Rio de Janeiro, Brazil

**Keywords:** Amazonian fruit, bioactive compounds, carotenoids, fruits out of the trade balance

## Abstract

Epidemiological studies have shown that a diet rich in bioactive components significantly reduces cardiovascular disease incidence and mortality. In this sense, there is a need for meta-analytical research that confirms this phenomenon and increases specific knowledge about certain bioactive compounds such as carotenoids. Thus, this systematic review and meta-analysis aim to disseminate knowledge about the sources of carotenoids in fruit consumed in the north of Brazil which are outside the Brazilian trade balance. A systematic review and a meta-analysis following the PRISMA guidelines were conducted based on a random effects synthesis of multivariable-adjusted relative risks (RRs). Searches of seven sources were carried out, including PubMed, Science Direct from Elsevier, Web of Science, Scielo, Eric Research and Google Scholar databases. The systematic review was guided by a systematic review protocol based on the POT strategy (population, outcome and type of study) adapted for use in this research. Mendeley was a resource used to organize and manage references and exclude duplicates of studies selected for review. In this review, we present the potential bioactive compounds concentrated in little-known fruit species from the Amazon and their benefits. Consuming fruits that are rich in notable constituents such as carotenoids is important for the prevention of chronic non-communicable diseases through anti-inflammatory and anticoagulant properties, as well as antivirals, immunomodulators and antioxidants agents that directly affect the immune response.

## 1. Introduction

Epidemiological studies have shown that there will be an increase in life expectancy, as well as a significant decrease in the incidence and prevalence of several inflammatory-based chronic diseases, such as cardiovascular diseases (CVDs). Among the factors that are related to this phenomenon are healthy eating habits, such as those based on the high consumption of foods rich in bioactive substances (natural products derived from plants, marine organisms and animals) [1,2]. In this sense, it is possible that there is a relationship between cardiovascular health and healthy eating, since oxidative stress may be related to CVDs and diets rich in fiber and bioactive compounds, mainly antioxidants (α-tocopherols, vitamin C and polyphenols) [3].

Furthermore, bioactive compounds have abundant inflammatory-based biological effects, going far beyond CVDs, including antitumor, anti-inflammatory, anticarcinogenic, antiviral, antimicrobial, antidiarrheal, antioxidant and other effects [4]. From this aspect, it is recognized that fruits and vegetables are important to healthy eating because they are made up of micronutrients, fibers and bioactive compounds [5,6,7]. However, according to the Household Budget Survey (POF) of the Brazilian Institute of Geography and Statistics (IBGE) regarding the population’s diet, sources of bioactive constituents such as vegetables and fruits, including juices, in the diet of the Brazilian population represent, on average, just over 5% of the calories ingested daily, a value lower than the recommendation from the World Health Organization (WHO) of 6–7% (400 g) for the consumption of these foods [8].

Studies have shown that food insecurity and/or low levels of consumption of bioactive compounds act as coadjutants in the development of chronic non-communicable diseases, ratified by data from statistical bodies. Thus, there is a paradox, as Brazil has vast food resources, such as foods rich in bioactive compounds, with an emphasis on commercialized fruits and non-commercialized fruits (fruit outside the export trade balance is called non-commercial) in the region [9,10].

In this research, we will focus on carotenoids, one of the main bioactive compounds present in fruits from the Brazilian Amazon region [11,12]. Carotenoids provide provitamin A and antioxidant, anti-inflammatory, neuroprotective, anticancer, cardioprotective, vision protection, photoprotective and immunomodulatory actions and are lipid deposit and storage modulators. In this way, they help in the prevention and treatment of various diseases [13,14,15,16]. In this sense, the action of carotenoids on the immune system stands out; this bioactive compound enables a significant improvement in the phagocytic and microbicidal capacity of neutrophils, in the concentration of intracellular calcium and in the production of nitric oxide. These activities are associated with decreased levels of superoxide anions, hydrogen peroxide, interleukin-6 (IL-6) and tumor necrosis factor alpha (TNF-α) [14,15].

Furthermore, they are responsible for the significant increases in lymphocyte proliferation and cytotoxic activity in natural killer (NK) cells; for the improvement in the production of immunoglobulin (IgM, IgA, IgG); for the activation of factor 2-related erythroid nuclear factor 2 (Nrf2); for reducing biomarkers of DNA damage and acute phase proteins, such as C-reactive protein (CRP); and for suppressing the activation of nuclear factor kappa B (NF-κB) mediated by hydrogen peroxide [14,15,16].

In addition, many of the aforementioned effects on the immune system are related to the fact that carotenoids act as exogenous antioxidants by optimizing the redox balance, neutralizing reactive oxygen species, playing a key role in the oxidative balance and protecting target structures and molecules from damage. In this context, they act as singlet molecular oxygen suppressors, convert hydroperoxides into more stable compounds, prevent the formation of free radicals by blocking oxidation reactions and inhibiting the autoxidation reaction chain and chelate pro-oxidant metals [15,16].

Also noteworthy is the provitamin A activity of some carotenoids, because, when converted into vitamin A, they become responsible for the formation and maturation of epithelial cells; promote proliferation; regulate thymocyte apoptosis; participate in the regulation of the differentiation, maturation and function of cells of the innate immune system; induce the migration of T cells to inflamed areas; act as a control and maintainer of regulatory T cell homeostasis; inhibit the development of inflammatory cells; exert effects on immunoglobulin production; and regulate B cell activity [14].

Given the above, this systematic review and meta-analysis aims to disseminate knowledge about the carotenoids in fruits consumed in the Brazilian Amazon rainforest region which are outside the Brazilian trade balance.

## 2. Methods

A systematic review and a meta-analysis were conducted following the PRISMA guidelines [17]. Three reviewers independently assessed the methodological quality of the included studies; there were no disagreements between reviewers. The revised Cochrane risk of bias tools for randomized (RoB-2) and non-randomized (Rob-1) trials was used to assess the quality of the included studies. Searches of six sources were carried out, including the PubMed, Science Direct from Elsevier, Web of Science, Scielo, Eric Research and Google Scholar databases. The systematic review was guided by a systematic review protocol based on the POT strategy (population, outcome and type of study) adapted for use in this research. Mendeley was a resource used to organize and manage references and exclude duplicates of studies selected for review. Sources with a time cutoff from 2016 to 2022 in any study area were considered.

The review was designed to highlight the benefits to human health from the presence of carotenoids in fruits and products consumed in the northern region of Brazil and outside the Brazilian export trade balance. In order to achieve the objective, the POT strategy protocol was used to summarize and analyze studies already carried out on the topic in question.

P—Population: Research that addressed the presence of carotenoids in fruits consumed in the northern region of Brazil. The studies included presented fruits outside the Brazilian export trade balance, excluding studies on animals, children or adolescents and in vitro studies, case reports, reviews, descriptive studies, opinion articles, technical articles, editorials, letters to the editor, theses, dissertations, publications at events, books and book chapters.

O—Outcome: Any type of study indicating benefits to human health from the consumption of fruits or products derived from them, such as the prevention of cardiovascular diseases, anti-inflammatory action, and presence of antioxidants and treatment of degenerative diseases.

T—Type of research: Research that addressed the presence of carotenoids in fruits outside the Brazilian export trade balance, indexed in the databases PubMed, Scopus, Science Direct from Elsevier, Web of Science, Latin American and Caribbean Literature in Health Sciences (Lilacs) and Google Scholar, in the form of scientific articles.

Three authors independently carried out the systematic searches in online databases, PubMed, Science Direct from Elsevier, Web of Science, Scielo, Eric Research and Google Scholar by using a search strategy contained by MeSH and free terms. The first descriptors used to build the search strategies were “bioactive compounds”, “Carotenoids”, “antioxidants”, “Amazonian fruits”, “nutraceuticals” and “functional foods”. Various combinations of descriptors were made using the Boolean operators AND and OR, taking into account the syntax rules of each base. These procedures were implemented in an effort to shorten the selection path.

The retrieved studies were transmitted to the EndNote X9TM software (ClarivateTM Analytics, Philadelphia, PA, USA), where duplicates were automatically excluded, and the remainder were removed manually. Gray literature was assessed manually in Microsoft WordTM 2010 (MicrosoftTM Ltd., Redmond, WA, USA).

### 2.1. Study Selection Process

A calibration exercise was carried out by three reviewers before beginning the study selection process, in which eligibility criteria were reviewed and applied to 20% samples of retrieved studies to determine inter-rater agreement. The selection process began once the appropriate degree of agreement (Kappa ≥ 0.81) was reached. Thus, the reviewers chose the studies after reading the titles and abstracts, with no disagreements between the examiners. Subsequently, eligible preliminary studies were obtained and fully evaluated.

### 2.2. Data Extraction

The data extracted from the articles that met the inclusion criteria were analyzed using a Discursive Textual Analysis approach [18], and, after fully reading all studies, the essential data chosen were extracted independently and blinded by reviewers.

## 3. Results

A total of 513 records were found in the investigated databases. Of these, 314 duplicates were excluded, leaving 199 articles. After reading the titles and abstracts, 152 studies that did not deal with carotenoids were excluded, leaving 47 for full text reading. Finally, 28 studies were excluded because their object of work was included in the Brazilian import trade list [19]. Therefore, only 19 articles were used to continue with the analyses. Figure 1 demonstrates the study selection process in detail following the PRISMA protocol.

Through the articles’ reading, it was possible classify and group the products as those that studied fresh fruits (9 papers), fruit-derived products: flour and oil (7 papers), and research using parts of fruits (7 papers); some works appeared in more than one category (4 papers). The contents of these research will be detailed in following Items 5, 6 and 7 and their categories are presented after a brief introduction of food compounds, bioactive compounds and carotenoids.

## 4. Properties of Some Food Compounds in the Prevention of Etiological Diseases

It is well known that foods such as vegetables, fruits, whole grains, legumes, nuts and fish display in their composition a diverse range of bioactive substances. These substances, phytochemicals, acids polyunsaturated fatty acids, vitamins, minerals and dietary fiber, are capable of reducing, inhibiting or delaying the onset and/or complications arising from inflammatory-based chronic non-communicable diseases. Although the mechanisms of protective effects by which these foods exert their effects are not entirely elucidated, there is some evidence of their anti-inflammatory and antioxidant effects [20,21,22,23,24,25,26].

Among the advantageous traits of these elements, one noteworthy aspect is their antioxidant capabilities. This feature allows them to directly counteract the harmful effects of free radicals or indirectly engage with enzymatic mechanisms that serve to protect against oxidative damage, thereby lowering the likelihood of cardiovascular diseases by safeguarding the endothelium. The bioactive compounds found in fruits and vegetables offer protection to the endothelium through a variety of mechanisms, including enhanced eNOS/NO bioavailability, a reduction in oxidative stress, suppression of the NF-κB pathway and decreased expression of cell adhesion molecules [27].

An appropriate diet rich in a variety of fruits, vegetables and legumes plays an active and protective role in supporting bodily functions. These foods serve as an abundant reservoir of nutritional resources, offering a wide array of bioactive compounds such as phenolic compounds, vitamins, carotenoids and minerals. Furthermore, they are recognized as valuable sources of soluble and insoluble dietary fiber, which play a vital role in overall nutrition [27,28,29,30].

Consumers are showing a growing interest in foods that offer more benefits than typical fare, including disease prevention and symptom reduction [29]. This heightened interest in functional foods, paired with advancements in technology, research and specific legislative regulations, has driven the food industry to extract functional compounds from fresh produce as well as agro-industrial by-products. This allows for the enrichment or introduction of food products with functional claims.

## 5. Bioactive Compounds

Foods are made up of nutrients and bioactive compounds, the latter can exist naturally or be added in the form of functional ingredients in the development of novel food products [7,31]. Bioactive compounds are not synthesized by humans and, for this reason, are not considered nutrients, since the growth and maintenance of the human body does not depend on their intake. Currently, there are no recommended daily intake values of them [23,32].

Bioactive compounds can be from three main sources: plants (fruits, vegetables, tubers, roots, cereals, legumes, bark, leaf, seeds and other plant parts), animals (kidney and liver) and microorganisms (Gram-positive bacteria, lactic acid from lactic acid bacteria and short-chain fatty acids provided by microorganisms) [33]. Bao et al. [34] state that these compounds have been shown to be of great help in maintaining health and reducing the risk of diseases due to their functional properties. Although the mechanisms of protective effects by which these foods exert their effects are not entirely elucidated, there is some evidence of the beneficial effects on human health, as shown in Table 1.

Phytochemicals are also known as bioactive compounds from plants. Considering a vast set of these compounds, Morand and Tomás-Barberán [35] proposed to divide them into categories: polyphenols, carotenoids, sulfur compounds, alkaloids and phytosterols, while Samtiya, Aluko and Moreno-rojas, [36] presented as a brief overview of bioactive compounds derived from plants: polyphenols, dietary fiber, carotenoids, vitamins, bioactive peptides and biogenic amines. Frequent consumption of these food sources, as well as direct supplementation of these compounds, offers several health benefits, including anti-aging effects, protection against cardiovascular disease, control and prevention of metabolic diseases and prevention and treatment of cancer, as well as protection against neurodegenerative diseases [36].

In their research, Ortega and Campos [37] outline diverse action mechanisms responsible for the positive health impacts of phytochemicals. These include the regulation of oxidative processes, enhanced expression of antioxidant enzymes, decreased synthesis of pro-inflammatory cytokines, heightened expression of neurotrophic factors and involvement in diverse intracellular signaling pathways. Despite these, there is a variety of other evidence of the beneficial health effects of bioactive compounds in the literature. The daily intake of bioactive rich-food sources acts favorably in the inflammatory body process and may be an efficient way to reduce the risk of NCDs, such as cardiovascular disease.

## 6. Carotenoids

Carotenoids are a class of tetraterpenoid pigments (C40) or terpenes with wide distribution in nature. They have different chemical structures and are represented by the formula (C_5_H_8_)n—with an extensive system of conjugated double bonds, which is called a chromophore, responsible for their characteristic colors (red, orange and yellow tones), properties and even special functions, such as photoprotection and photosynthesis [38,39]. More than 1000 natural carotenoids have been described, but only 40–50 are consumed in human diets, mainly: β-carotene, α-carotene, β-cryptoxanthin, lycopene, lutein and zeaxanthin.

Carotenoids are classified into carotenes and xanthophylls. The former are only composed by carbon and hydrogen, while the other group also contains oxygen [28,40]. Additionally, animals are unable to biosynthesize carotenoids, contrary to plants, algae, fungi, yeasts and bacteria. Vegetables constantly synthesize these substances, and thus their composition in edible vegetables can vary (Figure 2).

Some fruits may be source of pro-vitamin A, such as β-carotene, β-cryptoxanthin, α-carotene and β-zeacarotene, while others display outstanding antioxidant action (capacity to suppress reactive oxygen species and deactivate free radicals), such as zeaxanthin, violaxanthin, neoxanthin and phytoene [40]. Particularly, both β-carotene and lycopene, are known by their capacity to scavenge simple molecular oxygen and peroxyl radicals and therefore have beneficial effects to humans attributed to their role in protecting against oxidative processes [41]. Due to these properties, these substances are linked to the reduced risk of developing degenerative diseases such as cancer, cardiovascular diseases and cataract formation [4]. Therefore, studies on the composition of carotenoids in tropical fruits, such as those from Amazon rainforest region, are important [41,42]. Information on the carotenoid composition of some Amazonian fruits, their carotenoids profile and health benefits are shown in Table 2.

Some carotenoids-rich Amazonian fruits can be highlighted, such as buriti (*Mauritia vinifera*), peach palm (*Bactrys gasipaes*), tucumã-do-amazonas (*A. aculeatum*), bacuri/ouricuri (*Scheelea phalerata*) and umari (*Poraqueiba sericea*) [40]. Among them, *Mauritia flexuosa* is known for the highest concentration of β-carotene. Furthermore, some palm fruits display a high availability of biolipids that can stimulate the intestinal absorption of carotenoids, contributing to anti-inflammatory and hypoglycemic effects [40].

The diversity of carotenoids found in vegetables can be found associated with structures such as dietary fibers and other polysaccharides; thus, they need to be released prior to absorption. Giuntini et al. [32] suggest that the best use of carotenoids by the human body occurs through the processes of cooking food, as well as chewing and gastric hydrolysis, though the process is not yet elucidated. It is worth mentioning, that these compounds are sensitive to environment and oxidative factors, e.g., heat, light, oxygen, acids, among others [28]. The interest in carotenoids by the food industry has increased over the past years, not only due to their functional and health claims, but also because they can be used as natural food coloring agents to replace synthetic ones. This demand has happen also linked to consumers expectation on eating healthier and naturally formulated foods.

Studies that investigated the content of these pigments in Amazonian fruits, as well as in their derived products and by-products (agro-industrial residues) are still scarce, considering the high demand for natural coloring substances by food industries and other commercial segments. In this sense, the investigation of efficient methods to isolate and quantify carotenoids from Amazonian sources, considering their thermo-sensitivity, is needed. 

## 7. Research on Products Derived from Fruits Rich in Carotenoids: Flour and Oil

The Brazilian fruit market has been growing due to the high demand from consumers for healthy foods, rich in bioactive compounds, polyphenols, β-carotene and lycopene. Some tropical fruits have been traditionally used to diverse culinary preparations and industrial products, e.g., juices, jellies, sweets, among others [47,48]. It is known that agro-industrial processes are responsible for generating high amounts of solid residues, such as husks, bagasse and seeds. These organic materials display a diverse content of macromolecules, e.g., lipids, fibers, proteins and polysaccharides, which have great potential for its isolation and use in developing high-value products for different industries, e.g., food, dermo-cosmetic, pharmaceutical and biofuel industries [47,48].

Some by-products from Amazonian fruits have been studied over the years, and some works from 2017 to 2021 are shown in Table 3. These studies that investigated the total content of carotenoids (CT) in derived products (flour and oil) from the pulp of Amazonian fruits. 

The pulp flour that presented the highest carotenoids content was inajá, both ripe and unripe, followed by araçá-boi, while in buriti it was not detected. Inajá (*Maximiliana maripa*) is a palm tree native to the Amazon rainforest region that produces edible fruits, which can be consumed as cooked or uncooked fruits and also as a fermented beverage (wine-like). In the research from Barbi et al. [54], it was possible to observe the potential of the pulp flour of this fruit. The defatted flour obtained from this mature fruit pulp presented a higher content of total carotenoids (125.110 μg 100 g^−1^), which was associated with its intense visual orange color. Moreover, a lower amount of CT (42.290 μg 100 g^−1^) was found in defatted flours that displayed a light beige-green color [54]. Both mature and immature inajá flours had higher contents of CT than pequi (*Caryocar brasiliense*) flour (2.110–3.490 μg 100 g^−1^) [57].

Bernardina et al. [50] investigated the nutritional properties of an oven-dried araçá-boi (*Eugenia stipitata*) pulp flour. The research found a total carotenoids content close to 12.000 μg 100 g^−1^, which highlights its potential to enrich dietary intake, as well as the development of novel products based on it. It is worth mentioning that zeaxanthin, 15-cis-β-carotene and all-trans-α-carotene were found in ripe araçá-boi pulp, with 45% bioaccessibility [50]. The pulp from buriti fruit, *Mauritia flexuosa*, is also an important Amazonian source of carotenoids. However, its defatted flour did not show these compounds, which is expected, since they are fat-soluble substances and are extracted together with the lipidic materials during solid–liquid extraction processes, such as by using Soxhlet apparatus [58].

Amazonian palm fruits are recognized by their high lipid content, especially in their mesocarp, and some of them are also a rich source of carotenoids, as shown in Table 2. 

Inajá, *Maximiliana maripa*, is an important Amazonian fruit source of carotenoids, as well as their derived flours [55]. The pulp oils from peach palm, *Bactris gasipaes*, and Inajá, *Maximiliana maripa*, showed a TC content ranging from 832.4 μg 100 g^−1^ to 140.990 μg 100 g^−1^, respectively. The content of carotenoids is not only influenced by the species in question, but also the extraction methods used. 

The peach palm oil was obtained by Soxhlet extraction, in the work of Santos et al. [56], and displayed up to 832.4 ± 0.64 µg 100 g^−1^ of β-carotene, while the palm oil from *Elaeis guineensis*, extracted using the same methodology, showed only 25.26 ± 0.82 mg g^−1^ of β-carotene [59]. These differences highlight the potential of peach palm fruit as a vegetable source of edible oils. It is worth saying that *Elaeis guineensis* is frequently used as a source of vegetable oil in diverse applications. In another work carried out by Santos et al. [51], the oven and freeze-dried bacaba (*Oenocarpus bacaba*) flour showed a total CT ranging from 908.17 μg 100 g^−1^ to 1068.3 μg 100 g^−1^, respectively. This difference was attributed to the drying conditions since carotenoids are known for being thermosensitive. In the study carried out by Mesquita et al. [53] with commercial buriti oil samples, the total carotenoids content ranged from 83.691 to 103.696 μg 100 g^−1^. When the commercial buriti oil samples were submitted to enzymatic interesterification, the CT content doubled to around 2786.83 ± 113.09 µg g^−1^ [53]. The authors explained that the lower CT content happened because of the analysis processes, especially because of the formation of secondary compounds (epoxycarotenes and apocarotenes) which are linked to oxidation and isomerization reactions due to exposure oxidative environmental factors, resulting in decreased carotenoids levels [60]. This behavior could also be explained by the antioxidant properties of β-carotenes, which can oxidate itself under oxidative damage and prevent lipid oxidation in buriti oil [61].

The inajá oils obtained by Soxhlet extraction (with ethanol) from unripe and ripe pulp flour showed a CT content ranging from 76.210 ± 0.21 to 96.980 ± 0.20 μg 100 g^−1^. As expected, the oil from the ripe pulp flour displayed the highest carotenoids content [55]. In the work of Barbi et al. [54], Inajá pulp oil obtained with subcritical propane displayed a carotenoids content of 140.99 mg 100 g^−1^. When the same extraction procedure was used to obtain oil from pumpkin residues, Cuco et al. [62] found higher levels of carotenoids in the peel (1655.40 ± 24.90 mg 100 g^−1^) than in the seeds (3.62 ± 0.48 mg 100 g^−1^). 

Therefore, derived products from Amazonian fruits have high nutritional, functional and technological potential for diverse applications. Nevertheless, in Brazil there are scarce industries aiming to obtain rich-carotenoid vegetable oils, especially from the Amazon rainforest region, because of diverse reasons that will not be presented in this article.

## 8. Research Using Parts of Rich-Carotenoids Fruits 

Over the past years, research has grown on the diverse use of agro-industrial residues all over the world and it is no different in Brazil. Materials from husks, seeds, endocarp and others are transformed into co-products after different processes, which can add a high value to them and to the industrial chain. These by-products can be a source of many macromolecules and bioactive substances, e.g., carotenoids. From 2017 to 2021, the published papers analyzed in this work addressed the carotenoids content from Amazonian fruit residues; however, few have presented their profile or the main class of carotenoids investigated. The content of carotenoids of some Amazonian fruit by-products are listed in Table 4.

Among the fruits presented in the Table 4, the peel of peach palm, tucumã and buriti stands out as having the highest content of carotenoids, ranging from around 33.000 to 21.000 μg 100 g^−1^, respectively. The amount of these substances can vary according to the extraction method and sample preparation, as well as cultivation conditions such as soil and climate. 

The achachairu (*Garcinia humilis*), araça-boi (*Eugenia stipitata*) and bacaba (*Oenocarpus bacaba*) are used in domestic culinary preparations as well as in industrial facilities to produce beverages, juices, jams and ice-cream. As far as it is known, there are scarce studies reporting the composition of bioactive compounds from these fruit residues. The research from Barros et al. [63] investigated the presence of bioactive compounds in by-products (seeds, peel and a very small amount of pulp) from different Amazonian fruits, *Garcinia humilis*, *Eugenia stipitata* and *Oenocarpus bacaba*, which were considered waste by food processing industries. The residues from araçá-boi displayed the highest content of total carotenoids, 3.339 ± 0.0 μg 100 g^−1^, compared with bacaba and achachairu [63]. Comparing the data content of carotenoids in araçá-boi peel from the work of Barros et al. [63] with those reported by Berni et al. [12], it was observed that the first found a much higher value than the second, 2.10 μg g^−1^, in araçá peel, and as well as in inajá pulp: 1371 μg 100 g^−1^ in the work of Anunciação et al. [45]. The residues from bacaba showed the value of 1.547 ± 0.01 μg 100 g^−1^, similar to that described by Santos et al. [51] for the freeze-dried bacaba pulp, 1068.30 ± 10.50 μg 100 g^−1^. The achachairu residue showed a higher value (932 ± 0.03 μg 100 g^−1^) of total carotenoids when compared to the muruci pulp (756.5 ± 20.5 μg 100 g^−1^) according to Belisário et al. [46]. These residues can be used as raw material for the extraction and isolation of carotenoids, which are highly necessary for the well-being and health of the humans.

Traditionally, buriti pulp is used to prepare sweet food products such as juices, jellies, ice cream, jams and even some flour. The industrial extraction of buriti oil generates a great amount of residues, which are composed by husks (~2500 tons/year), endocarp and cellulose bran (~6000 tons/year) [58]. It is important to highlight that the buriti endocarp is considered a residue not used in food formulations. Hence, more studies are necessary to fully understand its potentiality. The work from Cardoso et al. [58] verified that carotenoids are a current bioactive substance in buriti residues (bark and endocarp). Buriti bark showed the highest carotenoid content, 21.030 μg 100 g^−1^, compared to its endocarp, 6.050 μg 100 g^−1^, and both residues showed higher values than the pulp, 1209.20 ± 49.65 µg g^−1^ [58]. Bleached shell and endocarp flours had a lower total carotenoid content, 1040.1 ± 11.3 and 150.5 ± 38.5 μg 100 g^−1^, than the unbleached samples, 1186.7 ± 22.0 and 291.2 ± 17.3 μg 100 g^−1^, which could be related to leaching or degradation, as water bleaching requires more processing time, resulting in an increased loss of minerals and vitamins [52]. In another work, buriti bark flour displayed a lower carotenoids value (2.08 ± 0.06 mg 100 g^−1^) for both epicarp and endocarp [65]. This behavior could be related to the processing temperature used by Morais et al. [65], which may led to a decreased concentration of it.

Camu-camu is recognized for its high antioxidant capacity, due to its greatest content of vitamin C and total phenolics, which makes its pulp so appreciated in the northern region of Brazil. As a good volume of this fruit is used for industrial food production, a great amount of residues can be generated, since only the pulp is aimed to this end. Thus, studies that investigate the properties of its by-products (peel and seed) are very important. The research carried out by Souza et al. [44] investigated the effect of different maturation stages on the composition of camu-camu residues. The peel in the mature phase showed the highest carotenoid content, 10.588 μg 100 g^−1^, compared with the caranã (*Mauritia carana*) pulp, 912 μg 100 g^−1^, which is another Amazonian palm tree [45].

Amariz et al. [64] addressed the total carotenoids content in by-products of fruit processing at the pulp refining stage. Among the fruits involved their research, it is worth highlighting the Amazonian fruits, taperebá and cupuaçu, which had a total carotenoid content of 7.000 ± 4 μg 100 g^−1^ and 620 ± 0.7 μg 100 g^−1^, respectively [64].

Peels from tucumã and peach palm are agro-industrial residues and can be also considered a rich source of natural pigments and/or health-promoting bioactive compounds, such as carotenoids [49]. The carotenoid profiles of peels from tucumã and peach palm were studied by Matos et al. [49], who identified β-carotene (7.8 ± 2.0 mg 100 g^−1^ and 7.3 ± 0.4 mg 100 g^−1^) as the main carotenoid, followed by γ-carotene (0.8 ± 0.4 mg 100 g^−1^ 1.8 ± 22 mg 100 g^−1^) and δ-carotene (0.3 ± 0.1 mg 100 g^−1^ and 1.74 ± 0.04 mg 100 g^−1^). These compounds are important since they can show vitamin A activity. The peels from tucumã and peach palm fruits displayed 2.1 times more total carotenoids, 1.806 μg 100 g^−1^ and 3.369 μg 100 g^−1^, respectively, than their pulps [49]. In the study of Martínez-Girón et al. [66], the peach palm epicarp flour displayed 59.31 ± 1.6 mg 100 g^−1^ of total carotenoids content.

In the fruit processing industry, the manufacturing of candied fruits, jellies, fruit pastes, juices, pulps, nectars and beverages can be considered a secondary processing sector. Many studies have focused on the use of by-products derived from the primary transformation sector of Brazilian agriculture; however, the utilization of by-products from the whole industrial sector offers an opportunity to obtain many components, such as diverse bioactive substances, carbohydrates, pectins, fibers and pigments, among others [64]. Thus, the by-products (seeds, peel, endocarp and fibrous pulp material) of Amazonian fruits that are inadequately discarded by the primary and secondary sectors of the fruit industries have a high potential to be used as a source of carotenoids by the food, cosmetics and pharmaceutical industries.

## 9. Concluding Remarks

In this systematic review, the bioactive properties and the potential health benefits of carotenoids from Amazonian fruits were presented. The analysis was conducted following the PRISMA guidelines. Initially, 513 records were registered, but after some selection, only 19 articles were analyzed. These papers were classified onto three main groups according to the type of fruit part and derived materials. Up to 10 carotenoids-rich fruits from the Brazilian Amazon Rainforest region and their potential health benefits were discussed. Carotenoids play important role in the prevention and symptom alleviation of non-communicable chronic diseases, due to their anti-inflammatory, anticoagulant, antiviral, immunomodulatory and antioxidant properties, which can influence the immune response.

The presented fruits are well-known and consumed in the Amazonian region to different purposes, such as natural popular medicines, as artisanal cosmetics, in culinary recipes and others. Despite these common uses, there is scarce information about clinical studies aiming to explain the performance of these bioactive compounds in in vivo models. The consumer demand for healthier and naturally formulated foods has been driven, resulting in more studies with fruits and vegetables from the Amazon rainforest region. Additionally, the interest and use of residues from the fruit processing industries has grown, linked to a rising sustainability appeal, as well as to the opportunities from a circular economy. Peel, bark, seeds and defatted flour are some of the common residues from fruit processing industries. These materials are rich sources of high-value macromolecules and bioactive compounds, such as carotenoids. It worth mentioning that the whole use of fruits and vegetables can help the development of novel products in diverse applications and increase the added value to the chain industries, as well as to stimulate the bioeconomy of Brazil. 

The supporting literature of this work has highlighted the importance of carotenoids for human nutrition and the prevention of various non-communicable diseases, even with a direct role in the immune system or as an upholding post-treatment recovery therapy. Thus, the carotenoids from Amazonian fruits and derived products can be, in the near future, widely used in different biological, nutritional and health-related applications.

## Figures and Tables

**Figure 1 molecules-29-02190-f001:**
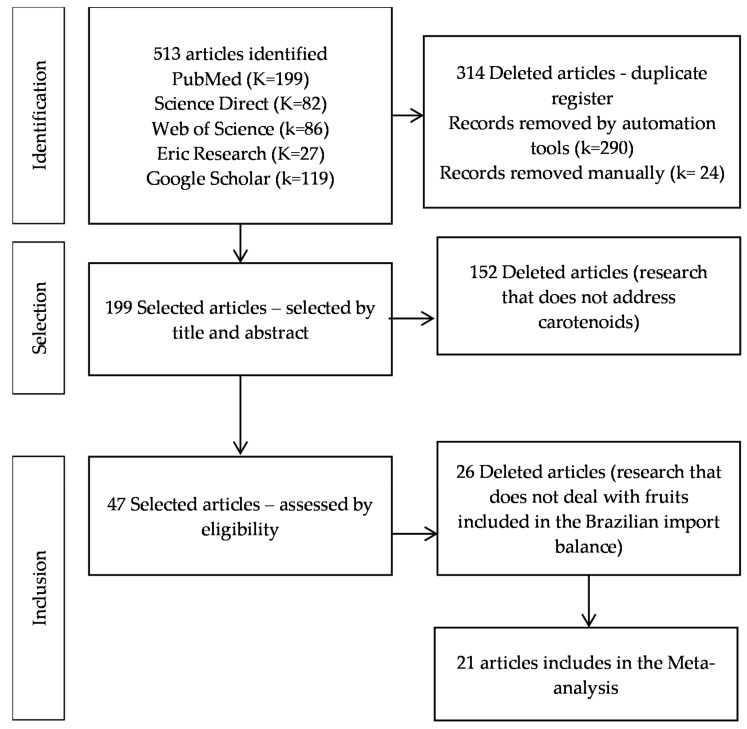
Flowchart of paper selection.

**Figure 2 molecules-29-02190-f002:**
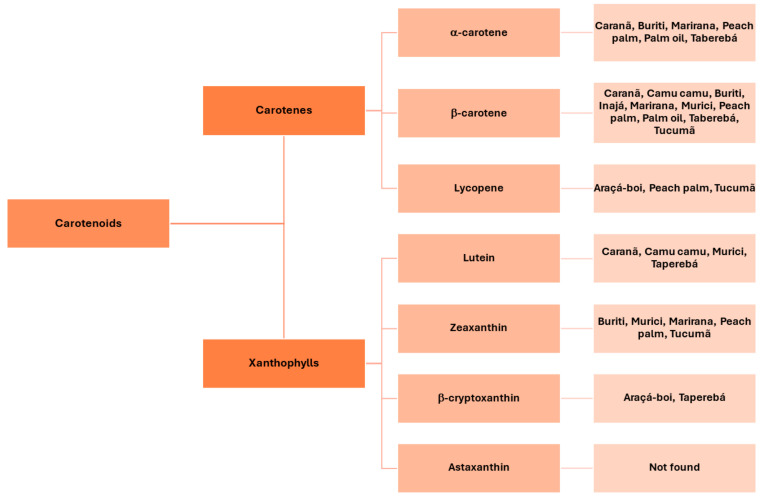
Classification of carotenoids and some Amazonian fruits (listed by their common names) presenting these compounds.

**Table 1 molecules-29-02190-t001:** Bioactive substances, subclasses, food sources and health benefits.

Bioactive Substance	Active Compounds	Biological Functions *	Foods
Phytochemicals	Phenolic compounds	Antioxidant activity; Anti-inflammatory; Contribute to balance and adequacy of intestinal functioning; Can help to reduce the absorption of fat and cholesterol.	Citrus fruits (lemon, orange and tangerine), in addition to other fruits such as cherry, grape, plum, pear, apple and papaya and vegetables (broccoli, red cabbage, onion, garlic and tomato), cereals, teas, coffee, cocoa, wine.
	Alkaloids	Acts mainly on the nervous system, whether central or autonomic.	Vegetable alkaloids are the main sources
	Organosulfur compounds	Prevention of cardiovascular diseases and the reduction in blood pressure, serum lipid level, blood glucose and oxidative stress.	Garlic, onion, chestnuts and walnuts.
	Carotenoids	Contribute to the body’s defenses, as they have antioxidant action, in addition to being responsible for the synthesis of vitamins, being related to reducing the risk of macular degeneration, cataracts and chronic diseases.	Dark green leafy vegetables (spinach), vegetables (carrots, peppers, lettuce, broccoli, among others) and tropical fruits, such as peach palm, tucumã (*Astrocaryum vulgare*), mango, taperebá (*Spondias mombin*), murici (*Byrsonima crassifolia*), guava, among others.
	Phytosterols	Help to reduce the absorption of cholesterol.	Vegetable oils (soybean and sunflower), fruits, seeds, leaves and stems.
Probiotics	Bifidobacteria and Lactobacilli	Improve balance of the intestinal microbiota; Benefits in the treatment of gastrointestinal diseases; Stimulation of the immune system.	Yogurts, fermented dairy products, kefir, kombucha and food supplements.
Prebiotics	Fibers, oligosaccharides, fructooligosaccharides and inulin	Collaborate for balance and adequacy function of intestines; Help to reduce the absorption of fat and cholesterol.	Fruits, oats, vegetables (chicory root and yacon potatoes), whole grains, tubers and bulbs, honey and brown sugar.
Polyunsaturated fatty acids	Omega 3 Omega 6	Reduction in LDL-cholesterol; Anti-inflammatory action; Indispensable for the development of brain and retina of newborns. Helps to adjust the triglyceride’ levels.	Vegetable oil (soybean, canola, wheat germ, flaxseed), nuts and marine fish (sardines, salmon, tuna, anchovies, herring).
Antioxidant vitamins	A	Antioxidant activity; Important in cell growth and differentiation; Preventive action in the development of tumors.	Animal products (liver, milk, eggs, butter, cheese and fish).
	C	Antioxidant activity; Decreased risk for certain types of cancer, cardiovascular disease and cataracts, as well as wound healing and immune modulation.	Fruits (orange, lemon, acerola, strawberry) and vegetables (broccoli, cabbage and spinach).
	E	Antioxidant activity; Anti-inflammatory; Adequacy of triglyceride levels.	Vegetable oils, wheat germ, oilseeds, dark green leafy vegetables and animal foods (egg yolks and liver).
Isoflavones and soy protein	Bioactive peptides	Hormonal regulation; Antioxidant activity; Cholesterol reduction.	Soy and derivatives

* As long as they are associated with a balanced diet and healthy living habits. Source: Henrique et al. [26]; Brasil [28]; Monsalve et al. [30].

**Table 2 molecules-29-02190-t002:** Some carotenoids-rich fruits from the Brazilian Amazon Rainforest region and their potential health benefits.

Popular Name	Scientific Name	Carotenoids Profile	Health Benefits	References
Araçá-boi 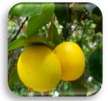	*Eugenia stipitata*	Lycopenes and cryptoxanthin	Antioxidant properties, anti-inflammatory and antidiabetic.	Araújo et al. (2019) [43]
Buriti 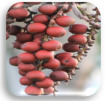	*Mauritia flexuosa Linn*. F	β-carotene	Reduce the incidence of xerophthalmia; Reducing the risk of developing cardiovascular disease.	Milanez et al. (2018) [41]; Neri-Numa et al. (2018) [42]
Camu-camu 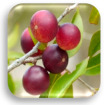	*Myrciaria dubia*	Trans-lutein and β-carotene	Antioxidant capacity.	Souza et al. (2018) [44]
Caranã 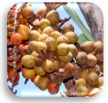	*Mauritiella armata*	Cis-β-carotene, trans -β-carotene,trans -α-carotene and trans-lutein	Lutein and zeaxanthin play an important role in reducing eye disorders due to their antioxidant, anti-inflammatory properties and ability to filter blue light.	Anunciação et al. (2019) [45]
Inajá 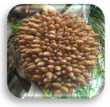	*Attalea maripa*	Trans -β-carotene	Antioxidant role, protects the cell body against the excess free radicals.	Anunciação et al. (2019) [45]
Murici 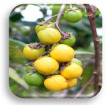	*Byrsonima crassifolia*	Lutein, zeaxanthin and β-carotene	High antioxidant potential; Inhibition of various degenerative processes; Source of Vitamin A.	Belisário et al. (2020) [46]
Marirana 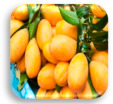	*Couepia subcordata Benth*	Trans -α-carotene, trans-β-carotene and zeaxanthin	May reduce the risk of macular degeneration and cataract formation.	Anunciação et al. (2019) [45]
Peach palm 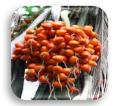	*Bactris gasipaes*	Zeaxanthin, α-carotene, β-carotene and lycopene.	Antioxidants, protecting the body against chronic diseases and certain cancers, macular degeneration, cataracts, neurological disorders, gastric anti-ulcer activity and strengthening the immune system.	Otero et al. (2020) [47]
Taperebá 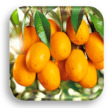	*Spondias mombin*	β-cryptoxanthin, α-cryptoxanthin, lutein, trans-α-carotene, cis-α-carotene and trans-β-carotene.	Strong potential in neurocognitive function together with a healthy lifestyle to promote brain health.	Assis et al. (2020) [48]
Tucumã 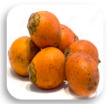	*Astrocaryum vulgare*	β-carotene, γ-carotene and δ-carotene	Source of pro-vitamin A.	Matos et al. (2019) [49]

**Table 3 molecules-29-02190-t003:** Carotenoids content in derived products (flour and oil) from different Amazonian fruits.

Popular Names	Scientific Names	Products	Methods	Total Carotenoids (μg/100 g)	References
Araçá-boi	*Eugenia stipitata*	Flour	Kiln-drying	12.000	Bernardina et al. (2020) [50]
Bacaba	*Oenocarpus bacaba*	Oil from pulp flour	Drying in freeze dryer	1068.30	Santos et al. (2021) [51]
		Oil from pulp flour	Kiln-drying	908.17	Santos et al. (2021) [51]
Buriti	*Mauritia flexuosa Linn. F*	Defatted flour	Kiln-drying and Soxhlet (hexane)	Not detected	Resende, Franca, and Oliveira (2019) [52]
		Oil	Not detected	103.696	Mesquita et al. (2020) [53]
Inajá	*Maximiliana maripa*	Defatted flour (mature)	Drying in freeze dryer	125.110	Barbi et al. (2020) [54]
		Defatted flour (green)	Drying in freeze dryer	42.290	Barbi et al. (2020) [54]
		Oil from pulp flour (ripe)	Soxhlet (ethanol)	96.980	Barbi et al. (2020) [54]
		Oil from pulp flour (green)	Soxhlet (ethanol)	76.210	Barbi et al. (2020) [54]
		Oil	Subcritical (propane)	140.990	Barbi et al. (2019) [55]
Peach palm	*Bactris gasipaes* Kunth	Oil	Soxhlet (Petroleum ether)	832.4	Santos et al. (2020) [56]

**Table 4 molecules-29-02190-t004:** Content of carotenoids of some Amazonian fruit by-products.

Popular Name	Scientific Name	Part of the Fruit	Carotenoids Profile	Total Carotenoids (μg 100 g^−1^)	References
Achachairu	*Garcinia humilis*	Seeds, peel and a very small amount of pulp	β-carotene	932	Barros et al. (2017) [63]
Araçá-boi	*Eugenia stipitata*	Seeds, peel and a very small amount of pulp	β-carotene	3.339	Barros et al. (2017) [63]
Bacaba	*Oenocarpus bacaba*	Seeds, peel and a very small amount of pulp	β-carotene	1.547	Barros et al. (2017) [63]
Buriti	*Mauritia flexuosa Linn. F*	Peel (epicarp)	β-carotene	21.030	Cardoso et al. (2020) [58]
		Bleached peel (epicarp)	β-carotene	1040.1	Resende et al. (2019) [52]
		Endocarp	β-carotene	6.050	Cardoso et al. (2020) [58]
		Bleached endocarp	β-carotene	150.5	Resende et al. (2019) [50]
Camu-camu	*Myrciaria dubia*	Peel	Trans-luteine β-carotene	10.588	Souza et al. (2018) [44]
Cupuaçu	*Theobroma grandiflorum*	Fibrous material from pulp and seeds	β-carotene γ-carotene and δ-carotene	620	Amariz et al. (2018) [64]
Peach palm	*Bactris gasipaes*	Peel		33.690	Matos et al. (2019) [49]
Taperebá	*Spondias mombin*	Fibrous material from pulp and seeds	Not determined	7.000	Amariz et al. (2018) [64]
Tucumã	*Astrocaryum vulgare*	Peel	β-carotene γ-carotene and δ-carotene	18.060	Matos et al. (2019) [49]

## Data Availability

Not applicable.

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
