# Peer review of "Sources of Carotenoids in Amazonian Fruits"

_molecules, 2024, doi:10.3390/molecules29102190_

Round 1

Reviewer 1 Report

Comments and Suggestions for Authors

The article is very innovative and meaningful, but the writing needs to be standardized. Please revise the yellow marks in the article carefully, including the format of references and figures. The investigation report of the author mainly refers to the literature of different databases, and if there is a chance,  can conduct research on relevant fruit plants in the later stage.

Comments on the Quality of English Language

English needs to be improved,especially the reference format.

Author Response

Thank you for reading and considering pertinent considerations for our article. All the points that you deatacou are adjusted (abbreviations, repeated words, references in the middle of the text). In particular, figure 1 was enlarged; A table (table 1) was inserted by recommendation from another evaluator, but rather addresses your questão about: What exactly are the ingredients in the food that makes the difference? You need to be specific, not general.

Reviewer 2 Report

Comments and Suggestions for Authors

The information presented in the manuscript it is relevant and gets readers to know more about the Brazilian fruits, and their health benefits and properties. I only have a question, aren´t any other fruits that contain poliphenols or flavonoids?

Author Response

Thank you for your willingness to evaluate our article and for your comments.

Regarding your question: aren't any other fruits that contain polyphenols or flavonoids? The answer is yes, but due to the search methodology used and the criteria chosen, these were the articles about fruits investigated.

Reviewer 3 Report

Comments and Suggestions for Authors

I really enjoyed reading this manuscript.

The manuscript entitled "Sources of carotenoid in Amazonian native fruit" is a review on the richness of bioactives in Amazonian native fruit. Is really well written and concerns a very interesting  subject.

The abstract focus on the main goal of the manuscript, highlights the main findings and on the positive impacts  of amazonian native fruits in human health when consumed.

The introduction is well focused on the relevance of carotenoids in human well-being, being responsible for several regulatory processes on the body.

Methods are described following a very simple approach, but I was expecting something more profound. It needs improvement.

The section about the effect of food compounds can be improved if a authors added a Figure containing all the information on the bioactive compounds effect on the body.

The section on bioactive compounds must be improved... it lacks deep.

Section 6 and 7 are really well written and a really good state-of-the art.

Author Response

Thank you for your consideration of our article and for your comments.
About your questions:

1. Methods are described following a very simple approach, but I was expecting something more profound. It needs improvement.

2. The section about the effect of food compounds can be improved if a authors added a Figure containing all the information on the bioactive compounds effect on the body.

3. The section on bioactive compounds must be improved... it lacks deep.

Thank you for your consideration of our article and for your comments.
About your questions:

We have the answer:

  1. 1. Adjustments were made to the figure describing the methodology that provides more clarification and a justification in the body of the text (before figure 1) indicating that details of the articles analyzed are found in the methodology.
    2. You requested a figure and we inserted a table (Table 1) which, in addition to highlighting the effects of bioactives on the body, also provides a certain level of depth on bioactives (referring to your question 3).
    3. The focus of the article is carotonoids, so to meet your third suggestion we detail in table 1 regarding the in-depth look at bioactives.

Reviewer 4 Report

Comments and Suggestions for Authors

I find the review “Sources of carotenoid in Amazonian native fruit” interesting. I am sending my observations:

Line 40 It  should be it

Line 157 What does mean NCDs ?

Line 167 What does mean NF-κB pathway?

Line 226 This paragraph is not understood very well. The figure does not show carotene synthesis routes. (Biosynthesis pathways, along with derivatives of the main components. The classification briefly shown in Figure 2.)  Please correct

Line 256 pol-ysaccharides, Please correct the separation of the word

Line 262  Please write this paragraph better.

The food industry's increased interest in carotenoids goes beyond functional and  health claims, as they can also be used as colorants. Hence the importance of investigating  the presence of these pigments in available foods, such as underused Amazonian fruits and even in agro-industrial residues. Therefore, the most selected methods for the analysis  of carotenoids in foods are: liquid chromatography and spectrophotometric analysis.

Line 278 I consider that this paragraph should not be. The next paragraph describes table 2

Line 280 Could you change the title of figure 2? Like carotene concentration of…………..

Line 292 with-out please check the word.

Line 321 concession-aire please check the word

Lines 342-345 In fruits mesocarp and pulp is the same please check the redaction of the paragraph

Line 354 Please change the redaction of the title of table 3

Line 450

Please check all the text in reference to the separation of words words at the end of lines.

In this review, we present the bioactive potentials and their benefits concentrated in little-known fruit species from the Amazon, especially substances that have adjuvant ac- tion in optimizing the human immune system. The bioactive constituents highlighted in this study were carotenoid have important effects for the prevention and treatment of non-communicable chronic diseases and their anti-inflammatory, anticoagulant, antiviral, im-munomodulatory and antioxidant effects that directly affect the response of immunolog- ical constituents.

These little-known elements of the Amazonian flora can be widely used in medi-cines, active ingredients in vaccines and other diverse dietary and pharmaceutical ap-pli cations. There is a need for further clinical studies aiming to explain in vivo models the performance of these bioactive compounds in cellular and animal models. The high- lighted approach of the therapeutic action in research already widely publicized brings the performance of constituents contained in these species of Amazonian fruit as an el-ement that can be applied in the near future in the prevention and recovery of various diseases, either as reinforcement constituents with preventive action on the immune sys-tem or post-treatment recovery in dietary therapies. Thus, the rise of these spe-cies is  based on the aggregation of value, mainly, given the ailments of hunger and malnutrition in the region, factors that reduce the immunological action.

Author Response

Thank you for the detailed considerations made to our article and for the comments.
About your questions:

We have the answer

  1. Line 40 It should be it: This has been corrected.
  2. Line 157 What does mean NCDs?: This has been corrected.
  3. Line 167 What does mean NF-κB pathway?: This has been corrected - nuclear factor kappa B (NF-κB).
  4. Line 226 This paragraph is not understood very well. The figure does not show carotene synthesis routes. (Biosynthesis pathways, along with derivatives of the main components. The classification briefly shown in Figure 2).  Please correct: This has been corrected.
  5. Line 256 pol-ysaccharides, Please correct the separation of the word: It was a formatting error.
  6. Line 262 Please write this paragraph better: This has been corrected.
  7. Line 278 I consider that this paragraph should not be. The next paragraph describes table 2: This has been corrected.
  8. Line 280 Could you change the title of figure 2? Like carotene concentration of………….. This has been corrected.
  9. Line 292 with-outplease check the word: It was a formatting error.
  10. Line 321 concession-aireplease check the word: It was a formatting error.
  11. Lines 342-345 In fruits mesocarp and pulp is the same please check the redaction of the paragraph: This has been corrected
  12. Line 354 Please change the redaction of the title of table 3: The title has been changed.
  13. Line 450: It was a formatting error.

Round 2

Reviewer 1 Report

Comments and Suggestions for Authors

After the author's modification of this article, the publication conditions of the compound journal

Comments on the Quality of English Language

English proficiency meets journal requirements

Reviewer 2 Report

Comments and Suggestions for Authors

The manuscript has been improved.